# Robust Generalized Method of Moments: A Finite Sample Viewpoint

**Dhruv Rohatgi**[*]
MIT

**Vasilis Syrgkanis**[†]
Stanford University

## Abstract

For many inference problems in statistics and econometrics, the unknown parameter is identified by a set of moment conditions. A generic method of solving moment conditions is the Generalized Method of Moments (GMM). However, classical GMM estimation is potentially very sensitive to outliers. Robustified GMM estimators have been developed in the past, but suffer from several drawbacks: computational intractability, poor dimension-dependence, and no quantitative recovery guarantees in the presence of a constant fraction of outliers. In this work, we develop the first computationally efficient GMM estimator (under intuitive assumptions) that can tolerate a constant $\epsilon$ fraction of adversarially corrupted samples, and that has an $\ell_2$ recovery guarantee of $O(\sqrt{\epsilon})$. To achieve this, we draw upon and extend a recent line of work on algorithmic robust statistics for related but simpler problems such as mean estimation, linear regression and stochastic optimization. As a special case, we apply our algorithm to instrumental variables linear regression with heterogeneous treatment effects, and experimentally demonstrate that it can tolerate as much as $10-15\%$ corruption, significantly improving upon baseline methods.

## 1 Introduction

Econometric and causal inference methodologies are increasingly being incorporated in automated large scale decision systems. Inevitably these systems need to deal with the plethora of practical issues that arise from automation. One important aspect is being able to deal with corrupted or irregular data, either due to poor data collection, the presence of outliers, or adversarial attacks by malicious agents. Even traditional applications of econometric methods, in social science studies, can greatly benefit from robust inference so as not to draw conclusions solely driven by a handful of samples, as was recently highlighted in [4].

One broad statistical framework, that encompasses the most widely used estimation techniques in econometrics and causal inference, is the framework of estimating models defined via *moment conditions*. In this paper we offer a robust estimation algorithm that extends prior recent work in robust statistics to this more general estimation setting.

For a family of distributions $\{\mathcal{D}_\theta : \theta \in \Theta\}$, identifying the parameter $\theta$ is often equivalent to solving

$$\mathbb{E}_{X \sim \mathcal{D}_\theta}[g(X, \theta)] = 0, \tag{1}$$

for an appropriate problem-specific vector-valued function $g$. This formalism encompasses such problems as linear regression (with covariates $X$, response $Y$, and moment $g((X,Y), \theta) = X(Y -$

---

[*]drohatgi@mit.edu. This work was partially done while the first author was an intern at Microsoft Research New England.

[†]vsyrgk@stanford.edu. This work was partially done while the second author was a Principal Researcher at Microsoft Research New England.

36th Conference on Neural Information Processing Systems (NeurIPS 2022).

$X^T\theta$)) and instrumental variables (IV) linear regression (with covariates $X$, response $Y$, instruments $Z$, and moment $g((X, Y, Z), \theta) = Z(Y - X^T\theta)$).

Under simple identifiability assumptions, moment conditions are statistically tractable, and can be solved by the *Generalized Method of Moments* (GMM) [16]. Given independent observations $X_1, \ldots, X_n \sim \mathcal{D}_\theta$, the (unweighted) GMM estimator is

$$\hat{\theta} = \operatorname*{argmin}_{\theta \in \Theta} \left\| \frac{1}{n} \sum_{i=1}^{n} g(X_i, \theta) \right\|_2^2.$$

Of course, for general functions $g$, finding $\hat{\theta}$ (the global minimizer of a potentially non-convex function) may be computationally intractable. Stronger assumptions imply that all approximate *local* minima of the above function are near the true parameter, in which case the GMM estimator is efficiently approximable. For instrumental variables (IV) linear regression, these assumptions follow from standard non-degeneracy assumptions.

Due to its flexibility, the GMM estimator is widely used in practice (along with heuristic variants, in models where it is computationally intractable) [29]. Unfortunately, like most other classical estimators in statistics, the GMM estimator suffers from a lack of robustness: a single outlier in the observations can arbitrarily corrupt the estimate.

**Robust statistics** Initiated by Tukey and Huber in the 1960s, robust statistics is a broad field studying estimators which have provable guarantees even in the presence of outliers [18]. Outliers can be modelled as samples from a heavy-tailed distribution, or even as adversarially and arbitrarily corrupted data. Classically, robustness of an estimator against arbitrary outliers is measured by breakdown point (the fraction of outliers which can be tolerated without causing the estimator to become unbounded [14]) and influence (the maximum change in the estimator under an infinitesimal fraction of outliers [15]). These metrics have spurred development and study of numerous statistical estimators which are often used in practice to mitigate the effect of outliers (e.g. Huber loss for mean estimation, linear regression, and other problems [17]). Problems such as robust *univariate* mean estimation are by now thoroughly understood [24, 22], and have statistically and computationally efficient estimators.

Unfortunately, in higher dimensions, there has long appeared to be a tradeoff between robustness and computational tractability; as a result, much of the literature on high-dimensional robust statistics has focused on statistical efficiency at the expense of computational feasibility [5, 23, 13]. While there is a rich literature on IV regression and GMM in the context of robust statistics, those works either present computationally intractable estimators [21, 12] or are robust in the sense of bounded influence [1, 27, 20] but not robust against arbitrary outliers. Until the last few years, most high-dimensional statistical problems lacked robust estimators satisfying the following basic properties [7]:

1. Computational tractability (i.e. evading the curse of dimensionality)
2. Robustness to a constant fraction of arbitrary outliers
3. Quantitative error guarantees without dimension dependence.

Recently, a line of work on *algorithmic* robust statistics has blossomed within the theoretical computer science community, with the aim of filling this gap in the high-dimensional statistics literature. Estimators with the above properties have been developed for various fundamental high-dimensional problems, including mean and covariance estimation [7, 9], linear regression [10, 3], and stochastic optimization [26, 8]. However, practitioners in econometrics and applied statistics often employ more sophisticated inference methods such as GMM and IV regression [29, 2]. Such methods are not traditionally under the purview of theoretical computer science and learning theory; perhaps as a result, computationally and statistically efficient robust estimators are still lacking.

**Our contribution** We address this lack. Methodologically speaking, our main contribution is to introduce GMM to the algorithmic robust statistics literature and vice versa (even aside from robustness, basic algorithmic questions about GMM remain open and surprisingly unstudied). Theoretically speaking, we prove that a simple modification to the SEVER algorithm for robust stochastic optimization [8] (based on using higher-derivative information) yields a computationally efficient and provably robust GMM estimator under intuitive deterministic assumptions about the uncorrupted

data. We instantiate this estimator for two important special cases of GMM—instrumental variables linear regression and instrumental variables logistic regression—under distributional assumptions about the covariates, instruments, and responses (and in fact our algorithm also applies to the IV generalized linear model under certain conditions on the link function).

Experimentally, we apply our algorithm to robustly solve IV linear regression. We find that it performs well for a wide range of instrument strengths. In the important setting of heterogeneous treatment effects, our algorithm tolerates as much as $10\%$ corruption. Applied to a seminal dataset previously used to estimate the effect of education on wages [6], we provide evidence for the robustness of the inference, and demonstrate that our algorithm can recover the original inference from corruptions of the dataset, significantly better than baseline approaches.

**Technical Overview**   Our robust GMM algorithm builds upon the SEVER algorithm and framework introduced in [8] for robust stochastic optimization, which itself builds on seminal work on robust multivariate mean estimation via spectral filtering [7, 9]. In this section, we outline the increasing levels of complexity.

First, given samples $v_1, \ldots, v_n \in \mathbb{R}^d$ among which $\epsilon n$ are corrupted, robust mean estimation asks for an estimate of the mean of the uncorrupted samples. The spectral filtering approach due to [9] iteratively does the following, until the sample covariance matrix is bounded: remove outliers in the direction of the largest variance. So long as the uncorrupted samples have bounded covariance, the filtering ensures that at termination, the empirical mean will approximate the uncorrupted mean.

Second, given functions $f_1, \ldots, f_n : \mathbb{R}^d \to \mathbb{R}$ among which $\epsilon n$ are corrupted, robust stochastic optimization asks for an approximate critical point of the mean of the uncorrupted functions. The SEVER algorithm [8] achieves this by alternating between (a) finding a critical point $\hat{w}$ of the current sample set $S$, and (b) applying one iteration of spectral filtering to the vectors $\{\nabla f_i(\hat{w}) : i \in S\}$, terminating when no samples are removed from $S$.[3] The termination guarantee of spectral filtering immediately implies that at termination, the average gradient of the uncorrupted samples at $\hat{w}$ is near the average gradient of the final sample set $S$ at $\hat{w}$, which is 0 by part (a). So $\hat{w}$ at termination is an approximate critical point of the mean of the uncorrupted functions.

In our problem, we are given functions $g_1, \ldots, g_n : \mathbb{R}^d \to \mathbb{R}^p$ among which $\epsilon n$ are corrupted, and wish to find an approximate minimizer of $\left\| \frac{1}{|U|} \sum_{i \in U} g_i(w) \right\|_2^2$, where $U \subseteq [n]$ is the set of uncorrupted functions. The obvious approach is to alternate between (a) finding a minimizer $\hat{w}$ of $\left\| \frac{1}{|S|} \sum_{i \in S} g_i(w) \right\|_2^2$, where $S$ is the current sample set, and (b) applying spectral filtering to the vectors $\{g_i(\hat{w}) : i \in S\}$, terminating when no samples are removed from $S$. The termination guarantee of spectral filtering implies that the final sample average $\frac{1}{|S|} \sum_{i \in S} g_i(\hat{w})$ is near the uncorrupted average $\frac{1}{|U|} \sum_{i \in U} g_i(\hat{w})$. Unfortunately, there is no guarantee that $\frac{1}{|S|} \sum_{i \in S} g_i(\hat{w})$ has small norm: part (a) only implies that $\hat{w}$ is a local minimizer (and hence critical point) of the norm, so

$$\frac{1}{|S|} \sum_{i \in S} (\nabla g_i(\hat{w}))^T \cdot \frac{1}{|S|} \sum_{i \in S} g_i(\hat{w}) = 0.$$

In the above equality, the sample gradient matrix at $\hat{w}$ could be arbitrarily corrupted, so the sample average at $\hat{w}$ could have arbitrarily large norm. In principle, even the *global* minimizer could have large norm. However, this issue can be fixed by using higher-derivative information: specifically, we also apply spectral filtering to (projections of) the matrices $\nabla g_i(\hat{w})$. Under appropriate boundedness and smoothness assumptions, it can then be shown that at termination (when neither filtering step removes samples), $\hat{w}$ is an approximate critical point of the norm of the uncorrupted average $\left\| \frac{1}{|U|} \sum_{i \in U} g_i(w) \right\|_2^2$. By a "strong identifiability" assumption, this implies that $\hat{w}$ is near the minimizer of $\left\| \frac{1}{|U|} \sum_{i \in U} g_i(x) \right\|_2^2$, as desired.

---

[3]A related approach simply applies robust mean estimation to estimate the gradients at each step of gradient descent [26].

## 2 Preliminaries

For real scalars or vectors $\{\xi_i\}_{i \in S}$ indexed by a set $S$, we use the notation $\mathbb{E}_S[\xi_i]$ for the sample expectation $\frac{1}{|S|} \sum_{i \in S} \xi_i$. Similarly, if $\xi_i$ are scalars, then we define the sample variance $\mathrm{Var}_S(\xi_i) = \mathbb{E}_S(\xi_i - \mathbb{E}_S \xi_i)^2$. If $\xi_i$ are vectors then we define the sample covariance matrix $\mathrm{Cov}_S(\xi_i) = \mathbb{E}_S(\xi_i - \mathbb{E}_S \xi_i)(\xi_i - \mathbb{E}_S \xi_i)^T$. A random vector $X$ is $(4, 2, \tau)$-hypercontractive if $\mathbb{E}(\langle X, u \rangle)^4 \leq \tau(\mathbb{E}(\langle X, u \rangle)^2)^2$ for all vectors $u$.

**Definition 2.1.** For a closed set $\mathcal{H}$, a function $f : \mathcal{H} \to \mathbb{R}$, and $\gamma > 0$, a $\gamma$-approximate critical point of $f$ (in $\mathcal{H}$) is some $x \in \mathcal{H}$ such that for any vector $v$ with $x + \delta v \in \mathcal{H}$ for arbitrarily small $\delta > 0$, it holds that $v \cdot \nabla f(x) \geq -\gamma \|v\|_2$.

**Definition 2.2.** For a closed set $\mathcal{H}$, a $\gamma$-approximate critical point oracle $\mathcal{L}_{\gamma, \mathcal{H}}$ is an algorithm which, given a differentiable function $f : \mathcal{H} \to \mathbb{R}$ returns a $\gamma$-approximate critical point of $f$.

**Definition 2.3.** The (unscaled) *logistic function* $G : \mathbb{R} \to \mathbb{R}$ is defined by $G(x) = 1/(1 + e^{-x})$.

**Outline** In Section 3, we describe the robust GMM problem, and we describe deterministic assumptions on a set of corrupted sample moments, under which we'll be able to efficiently estimate the parameter which makes the uncorrupted moments small. In Section 4, we describe a key subroutine of our robust GMM algorithm, which is commonly known in the literature as *filtering*. In Section 5, we describe the robust GMM algorithm and prove a recovery guarantee under the assumptions from Section 3. In Section 6, we apply this algorithm to instrumental variable linear and logistic regression, proving that under reasonable stochastic assumptions on the uncorrupted data, arbitrarily $\epsilon$-corrupted moments from these models satisfy the desired deterministic assumptions with high probability. Finally, in Section 7, we evaluate the performance of our algorithm on two corrupted datasets.

## 3 Robust GMM Model

In this section, we formalize the model in which we will provide a robust GMM algorithm. Classically, the goal of GMM estimation is to identify $\theta \in \Theta$ given data $X_1, \ldots, X_n \sim \mathcal{D}_\theta$, using the moment condition $\mathbb{E}_{X \sim \mathcal{D}_\theta}[g(X, \theta)] = 0$. We consider the added challenge of the $\epsilon$-*strong contamination model*, in which an adversary is allowed to inspect the data $X_1, \ldots, X_n$ and replace $\epsilon n$ samples with arbitrary data, before the algorithm is allowed to see the data. This corruption model encompasses most reasonable sources of outliers.

For our main theorem, we do not make stochastic assumptions about $\{\mathcal{D}_\theta : \theta \in \Theta\}$. Instead, we make deterministic assumptions about the empirical moments $g_i(\theta) := g(X_i, \theta)$ of the given data, which are *robust to $\epsilon$-strong contamination*. Concretely, we make the following assumption.

**Assumption 3.1.** Given differentiable moments $g_1, \ldots, g_n : \mathbb{R}^d \to \mathbb{R}^p$, a corruption parameter $\epsilon > 0$, well-conditionedness parameters $\lambda$ and $L$, a Lipschitzness parameter $L_g$, and a noise level parameter $\sigma^2$, there is a set $I_{\text{good}} \subseteq [n]$ with $|I_{\text{good}}| \geq (1 - \epsilon)n$ (the "uncorrupted samples"), a vector $w^* \in \mathbb{R}^d$ (the "true parameter"), and a radius $R_0 \geq \|w^*\|_2$ with the following properties:

- **Strong identifiability.** $\sigma_{\min}(\mathbb{E}_{I_{\text{good}}} \nabla g(w^*)) \geq \lambda$

- **Bounded-variance gradient.** $\mathbb{E}_{I_{\text{good}}}(u^T \nabla g(w^*) v)^2 \leq L^2$ for all unit-vectors $u \in \mathbb{R}^p$, $v \in \mathbb{R}^d$

- **Bounded-variance noise.** $\mathbb{E}_{I_{\text{good}}}(v \cdot g(w^*))^2 \leq \sigma^2 L$ for all unit vectors $v$

- **Well-specification.** $\left\| \mathbb{E}_{I_{\text{good}}} g(w^*) \right\|_2 \leq \sigma \sqrt{L \epsilon}$

- **Lipschitz gradient.** $\left\| \mathbb{E}_{I_{\text{good}}} \nabla g(w) - \mathbb{E}_{I_{\text{good}}} \nabla g(w^*) \right\|_{\text{op}} \leq L_g \|w - w^*\|_2$ for all $w \in B_{2R_0}(0)$

- **Stability of gradient.** $R_0 < \lambda/(9L_g)$.

Intuitively, Assumption 3.1 can be thought of as a condition on the uncorrupted samples, because if they satisfy the assumption with parameter $\epsilon_0$, then after $\epsilon$-strong contamination, the corrupted samples will still satisfy the assumption with parameter $\epsilon_0 + \epsilon$. Strong identifiability is needed for parameter recovery (even without corruption). Bounded-variance gradient is a technical condition which e.g. reduces to a 4th moment bound for IV regression. The third and fourth conditions ensure

that the data is approximately well-specified by the moment conditions. The fifth and sixth conditions hold trivially for IV linear regression; for non-linear moment problems, such as our logistic IV regression problem, this condition requires that the $\ell_2$-norm of the parameters be sufficiently small, such that the logits do not approach the flat region of the logistic function, a condition that is natural to avoid loss of gradient information and extreme propensities.

# 4   The FILTER Algorithm

In many robust statistics algorithms, an important subroutine is a *filtering* algorithm for robust mean estimation. In this section we describe a filtering algorithm used in numerous prior works, including e.g. [8, 9]. Given a set of vectors $\{\xi_i : i \in S\}$ and a threshold $M$, the algorithm returns a subset of $S$, by thresholding outliers in the direction of largest variance. Formally, see Algorithm 1.

---

**Algorithm 1** FILTER

---

1: **procedure** FILTER($\{\xi_i : i \in S\}, M$)
2:    $\hat{\xi} \leftarrow \mathbb{E}_S[\xi_i], \text{Cov}_S(\xi_i) = \mathbb{E}_S[(\xi_i - \hat{\xi})(\xi_i - \hat{\xi})^T]$
3:    $v \leftarrow$ largest eigenvector of $\text{Cov}_S(\xi_i)$
4:    $\tau_i \leftarrow (v \cdot (\xi_i - \hat{\xi}))^2$ for $i \in S$
5:    **if** $\frac{1}{|S|} \sum_{i \in S} \tau_i \leq 24M$ **then**
6:       **return** $S$
7:    **else**
8:       Sample $T \leftarrow \text{Unif}([0, \max \tau_i])$
9:       **return** $S \setminus \{i \in S : \tau_i > T\}$

---

This algorithm has two important properties. First, if it does not filter any samples, then the sample mean is provably stable, i.e. it cannot have been affected much by the corruptions, so long as the uncorrupted samples had bounded variance (proof in Appendix B.1).

**Lemma 4.1** (see e.g. [8, 9]). *Suppose that* FILTER *does not filter out any samples. Then*

$$\|\mathbb{E}_S \xi - \mathbb{E}_I \xi\|_2 \leq 3\sqrt{48}\sqrt{(M + \|\text{Cov}_I(\xi)\|_{op})\epsilon}$$

*for any $I \subseteq [n]$ and $\epsilon > 0$ such that $|S|, |I| \geq (1 - \epsilon)n$.*

Second, if the threshold is chosen appropriately (based on the variance of the uncorrupted samples), then the filtering step always in expectation removes at least as many corrupted samples as uncorrupted samples. Equivalently, the size of the symmetric difference between the current sample set and the uncorrupted samples (i.e. the number of corrupted samples in the current set plus the number of uncorrupted samples which have been filtered out of the current set) always decreases in expectation (proof in Appendix B.1.1).

**Lemma 4.2** (see e.g. [8, 9]). *Consider an execution of* FILTER *with sample set $S$ of size $|S| \geq 2n/3$, and vectors $\{\xi_i : i \in S\}$, and bound $M$. Let $S'$ be the sample set after this iteration's filtering. Let $I_{good} \subseteq [n]$ satisfy $|I_{good}| \geq (5/6)n$. Suppose that $\text{Cov}_{I_{good}}(\xi_i) \preceq MI$, then*

$$\mathbb{E}|S' \triangle I_{good}| \leq \mathbb{E}|S \triangle I_{good}|,$$

*where the expectation is over the random threshold, and $\triangle$ denotes symmetric difference.*

# 5   The ITERATED-GMM-SEVER Algorithm

In this section, we describe and analyze an algorithm ITERATED-GMM-SEVER for robustly solving moment conditions under Assumption 3.1. The key subroutine is the algorithm GMM-SEVER, which given an initial estimate $w_0$ and a radius $R$ such that the true parameter is contained in $B_R(w_0)$, returns a refined estimate $w$ such that (with large probability) the radius bound can be decreased by a constant factor. We assume access to an approximate constrained critical point oracle $\mathcal{L}$ (Definition 2.2), which can be efficiently implemented (for arbitrary smooth bounded functions) by gradient descent.

---

**Algorithm 2** GMM-SEVER

1: **procedure** GMM-SEVER($\mathcal{L}, \{g_1, \ldots, g_n\}, w_0, R, \gamma, L, \sigma$)
2:     $S \leftarrow [n]$
3:     **repeat**
4:         Compute a $\gamma$-approximate critical point $w \leftarrow \mathcal{L}_{\gamma, B_R(w_0)}(\|\mathbb{E}_S(g_i(\cdot))\|_2^2)$
5:         $u \leftarrow \mathbb{E}_S g_i(w)$
6:         $S' \leftarrow \text{FILTER}(\{\nabla g_i(w) \cdot u : i \in S\}, L^2 \|u\|_2^2)$
7:         **if** $S' \neq S$ **then**
8:             Set $S \leftarrow S'$ and return to line 4
9:         $S'' \leftarrow \text{FILTER}(\{g_i(w) : i \in S\}, \sigma^2 L + 4L^2 R^2)$
10:        **if** $S'' \neq S$ **then**
11:            Set $S \leftarrow S''$ and return to line 4
12:     **until** $S'' = S$
13:     **return** $(w, S)$

---

**Algorithm 3** AMPLIFIED-GMM-SEVER

1: **procedure** AMPLIFIED-GMM-SEVER($\mathcal{L}, \{g_1, \ldots, g_n\}, w_0, R, \gamma, \epsilon, L, \sigma, \delta$)
2:     $t \leftarrow 0$
3:     **repeat**
4:         $w, S \leftarrow$ GMM-SEVER($\mathcal{L}, \{g_1, \ldots, g_n\}, w_0, R, \gamma, L, \sigma$)
5:         $t \leftarrow t + 1$
6:     **until** $|S| \geq (1 - 11\epsilon)n$ **or** $(1/10)^t \leq \delta$
7:     **return** $w$

---

Like the algorithm SEVER [8], our algorithm GMM-SEVER alternates (a) finding a critical point of a function associated to the current samples, and (b) filtering out "outlier" samples. Unlike SEVER, the function we optimize is not simply an empirical mean over the samples, but rather the squared-norm of the sample moments. Moreover, we need two filtering steps: the moments as well as directional derivatives of the moments, in a carefully chosen direction. See Algorithm 2 for the complete description.

We will only prove a constant failure probability for GMM-SEVER. However, we will show that it can be amplified to an arbitrarily small failure probability $\delta$. We call the resulting algorithm AMPLIFIED-GMM-SEVER; see Algorithm 3. The algorithm ITERATED-GMM-SEVER then consists of iteratively calling AMPLIFIED-GMM-SEVER to refine the parameter estimate and bound the true parameter within successively smaller balls; see Algorithm 4.

We start by analyzing GMM-SEVER. In the next two lemmas, we show that if the algorithm does not filter out too many samples, then we can bound the distance from the output to $w^*$. First, we show a first-order criticality condition (in the direction $\hat{w} - w^*$) for the norm of the moments of the "good" samples. If there was no corruption, then we would have an inequality of the form

$$\frac{(\hat{w} - w^*)^T}{\|\hat{w} - w^*\|_2} \mathbb{E}_{I_{\text{good}}} \nabla g(\hat{w})^T \mathbb{E}_{I_{\text{good}}} g(\hat{w}) \leq \gamma.$$

With $\epsilon$-corruption, the algorithm is designed so that we can still show the following inequality, matching the above guarantee up to $O(\sqrt{\epsilon})$ (proof in Appendix C.1):

**Lemma 5.1.** *Suppose that the input parameters $R$ and $w_0$ satisfy $B_R(w_0) \subseteq B_{2R_0}(0)$. Under Assumption 3.1, at algorithm termination, if $|S| \geq (1 - 10\epsilon)n$, then the output $\hat{w}$ of GMM-SEVER satisfies*

$$\frac{(\hat{w} - w^*)^T}{\|\hat{w} - w^*\|_2} \mathbb{E}_{I_{good}} \nabla g(\hat{w})^T \mathbb{E}_{I_{good}} g(\hat{w}) \leq \gamma + 275\sigma L^{3/2}\sqrt{\epsilon} + 603L^2 R\sqrt{\epsilon}$$

Moreover, we can show that any point satisfying the first-order criticality condition must be close to $w^*$, using the least singular value bound on the gradient (proof in Appendix C.2).

**Lemma 5.2.** *Suppose that the input parameters $R$ and $w_0$ satisfy $B_R(w_0) \subseteq B_{2R_0}(0)$. Under Assumption 3.1, suppose that $w \in B_R(w_0)$ satisfies*

$$(w - w^*)^T \mathbb{E}_{I_{good}} \nabla g(w)^T \mathbb{E}_{I_{good}} g(w) \leq \kappa \|w - w^*\|_2 .$$

---
**Algorithm 4** ITERATED-GMM-SEVER
---
1: **procedure** ITERATED-GMM-SEVER($\{g_1, \ldots, g_n\}, R_0, \gamma, \epsilon, \lambda, L, \sigma, \delta$)
2: $\quad$ $t \leftarrow 1, w_1 \leftarrow 0, R_1 \leftarrow R_0, \delta' \leftarrow c\delta/\log(R\sqrt{L}/(\sigma\sqrt{\epsilon})), \gamma = \sigma L^{3/2}\sqrt{\epsilon}$
3: $\quad$ **repeat**
4: $\quad\quad$ $\hat{w}_t := \text{AMPLIFIED-GMM-SEVER}(\{g_1, \ldots, g_n\}, w_t, R_t, \epsilon, L, \sigma, \gamma, \delta')$
5: $\quad\quad$ $R_{t+1} \leftarrow 2\gamma/\lambda^2 + C((L^2/\lambda^2)R_t\sqrt{\epsilon} + \sigma(L^{3/2}/\lambda^2)\sqrt{\epsilon})$
6: $\quad\quad$ $t \leftarrow t + 1$
7: $\quad$ **until** $R_t > R_{t-1}/2$
8: $\quad$ **return** $\hat{w}_{t-1}$
---

*Then* $\|w - w^*\|_2 \leq 4(\kappa + \sigma L^{3/2}\sqrt{\epsilon})/\lambda^2$.

Putting the above lemmas together, we immediately get the following bound on $\|\hat{w} - w^*\|_2$.

**Lemma 5.3.** *Suppose that the input parameters $R$ and $w_0$ satisfy $B_R(w_0) \subseteq B_{2R_0}(0)$. Under Assumption 3.1, at algorithm termination, if $|S| \geq (1 - 10\epsilon)n$, then the output $\hat{w}$ of GMM-SEVER satisfies*

$$\|\hat{w} - w^*\|_2 \leq \frac{4\gamma}{\lambda^2} + 2412(L^2/\lambda^2)R\sqrt{\epsilon} + 1102\sigma(L^{3/2}/\lambda^2)\sqrt{\epsilon}.$$

It remains to bound the size of $S$ at termination. We follow the super-martingale argument from [8], which uses Lemma 4.2 (proof in Appendix C.3).

**Theorem 5.4.** *Suppose that the input parameters $R$ and $w_0$ satisfy $B_R(w_0) \subseteq B_{2R_0}(0)$. Let $\hat{w}$ be the output of GMM-SEVER. Then with probability at least $9/10$, it holds that*

$$\|\hat{w} - w^*\|_2 \leq \frac{4\gamma}{\lambda^2} + 2412(L^2/\lambda^2)R\sqrt{\epsilon} + 1102\sigma(L^{3/2}/\lambda^2)\sqrt{\epsilon}.$$

*The time complexity of GMM-SEVER is $O(\text{poly}(n, d, p, T_\gamma))$ where $T_\gamma$ is the time complexity of the $\gamma$-approximate learner $\mathcal{L}$. Moreover, for any $\delta > 0$ the success probability can be amplified to $1 - \delta$ by repeating GMM-SEVER $O(\log 1/\delta)$ times, or until $|S| \geq (1 - 10\epsilon)n$ at termination. We call this AMPLIFIED-GMM-SEVER, and it has time complexity $O(\text{poly}(n, d, p, T_\gamma) \cdot \log(1/\delta))$.*

With the above guarantee for GMM-SEVER and AMPLIFIED-GMM-SEVER, we can now analyze ITERATED-GMM-SEVER (proof in Appendix C.4).

**Theorem 5.5.** *Suppose that the input to ITERATED-GMM-SEVER consists of functions $g_1, \ldots, g_n$ : $\mathbb{R}^d \to \mathbb{R}^p$, a corruption parameter $\epsilon > 0$, well-conditionedness parameters $\lambda$ and $L$, a Lipschitzness parameter $L_g$, a noise level parameter $\sigma^2$, a radius bound $R_0$, and an optimization error parameter $\gamma$, such that Assumption 3.1 is satisfied for some unknown parameter $w^* \in \mathbb{R}^d$, and $(L^2/\lambda^2)\sqrt{\epsilon} \leq 1/9648$.* [4] *Suppose that the algorithm is also given a failure probability parameter $\delta > 0$.*

*Then the output $\hat{w}$ of ITERATED-GMM-SEVER satisfies*

$$\|\hat{w} - w^*\|_2 \leq O(\sigma(L^{3/2}/\lambda^2)\sqrt{\epsilon})$$

*with probability at least $1 - \delta$. Moreover, the algorithm has time complexity $O(\text{poly}(n, d, p, T_\gamma) \cdot \log(1/\delta) \cdot \log(R\sqrt{L}/(\sigma\sqrt{\epsilon})))$, where $T_\gamma$ is the time complexity of a $\gamma$-approximate learner and $\gamma = \sigma L^{3/2}\sqrt{\epsilon}$.*

## 6 Applications

In this section, we apply ITERATED-GMM-SEVER to solve linear and logistic instrumental variables regression in the strong contamination model.

---

[4]This constant may be improved; we focus in this paper on dependence on the parameters of the problem and do not optimize constants.

**Robust IV Linear Regression**  Let $Z$ be the vector of $p$ real-valued instruments, and let $X$ be the vector of $d$ real-valued covariates. Suppose that $Z$ and $X$ are mean-zero. Suppose that the response can be described as $Y = X^T w^* + \xi$ for some fixed $w^* \in \mathbb{R}^d$. The distributional assumptions we will make about $X$, $Y$, and $Z$ are described below.

**Assumption 6.1.**  Given a corruption parameter $\epsilon > 0$, well-conditionedness parameters $\lambda$ and $L$, hypercontractivity parameter $\tau$, noise level parameter $\sigma^2$, and norm bound $R_0$, we assume the following: (i) **Valid instruments:** $\mathbb{E}[\xi|Z] = 0$, (ii) **Bounded-variance noise:** $\mathbb{E}[\xi^2|Z] \leq \sigma^2$, (iii) **Strong instruments:** $\sigma_{\min}(\mathbb{E} Z X^T) \geq \lambda$, (iv) **Boundedness:** $\|\mathrm{Cov}([Z;X])\|_{\mathrm{op}} \leq L$, (v) **Hypercontractivity:** $[Z;X]$ is $(4, 2, \tau)$-hypercontractive, (vi) **Bounded 8th moments:** $\max_i X_i^8 \leq O(\tau^2 L^4)$ and $\max_i Z_i^8 \leq O(\tau^2 L^4)$ (vii) **Bounded norm parameter:** $\|w^*\|_2 \leq R_0$.

For intuition, conditions (i – iii) are standard for IV regression even in the absence of corruption; (iv – vi) are conditions on the moments of the distribution, and hold for a variety of reasonable distributions including but not limited to any multivariate Gaussian distribution with bounded-spectral-norm covariance. Condition (vii) essentially states that we need an initial estimate of $w^*$ (but the time complexity of our algorithm will depend only logarithmically on the initial estimate error $R_0$).

Define the random variable
$$g(w) = Z(Y - X^T w)$$
for $w \in \mathbb{R}^d$, and let $(X_i, Y_i, Z_i)$ be $n$ independent samples drawn according to $(X, Y, Z)$. Let $\epsilon > 0$. We prove that under the above assumption, if $n$ is sufficiently large, then with high probability, for any $\epsilon$-contamination $(X'_i, Y'_i, Z'_i)_{i=1}^n$ of $(X_i, Y_i, Z_i)_{i=1}^n$, the functions $g_i(w) = Z'_i(Y'_i - (X'_i)^T w)$ satisfy Assumption 3.1. Formally, we prove the following theorem (see Appendix D):

**Theorem 6.2.** *Let $\epsilon > 0$. Suppose that $\epsilon < c\min(\lambda^2/(\tau L^2), \lambda^4/L^4)$ for a sufficiently small constant $c > 0$, and suppose that $n \geq C(d + p)^5 \tau \log((p + d)/\tau\epsilon)/\epsilon^2$ for a sufficiently large constant $C$. Then with probability at least $0.95$ over the samples $(X_i, Y_i, Z_i)_{i=1}^n$, the following holds: for any $\epsilon$-corruption of the samples and any upper bound $R_0 \geq \|w^*\|_2$, Assumption 3.1 is satisfied. In that event, if $L$, $\lambda$, $\sigma$, and $\epsilon$ are known, then there is a $\mathrm{poly}(n, d, p, \log(1/\delta), \log(R_0/(\sigma\sqrt{\epsilon})))$-time algorithm which produces an estimate $\hat{w}$ satisfying $\|\hat{w} - w^*\|_2 \leq O(\sigma(L^{3/2}/\lambda^2)\sqrt{\epsilon})$ with probability at least $1 - \delta$.*

**Robust IV Logistic Regression**  Let $Z$ be a vector of $p$ real-valued instruments, and let $X$ be a vector of $d$ real-valued covariates. Suppose that $Z$ and $X$ are mean-zero. Suppose that the response can be described as $Y = G(X^T w^*) + \xi$ for some fixed $w^* \in \mathbb{R}^d$, where $G$ is the (unscaled) logistic function. The proofs only use 1-Lipschitzness of $G$ and $G'$, and that $G'(0)$ is bounded away from $0$.

As far as distributional assumptions, we assume in this section that Assumption 6.1 holds, and additionally assume that the norm bound satisfies $R_0 \leq c\min(\lambda^2/L, \lambda/\sqrt{\tau L^3})$ for an appropriate constant $c$, where $\lambda$, $L$, and $\tau$ are as required for the Assumption. We obtain the following algorithmic result (proof in Appendix E):

**Theorem 6.3.** *Let $\epsilon > 0$. Suppose that $\epsilon < c\min(\lambda^2/(\tau L^2), \lambda^4/L^4)$ for a sufficiently small constant $c > 0$, and suppose that $n \geq C(d + p)^5 \tau \log((p + d)/\tau\epsilon)/\epsilon^2$ for a sufficiently large constant $C$. Suppose that $\|w^*\|_2 \leq R_0 \leq c\min(\lambda^2/L, \lambda/\sqrt{\tau L^3})$. Then with probability at least $0.95$ over the samples $(X_i, Y_i, Z_i)_{i=1}^n$, the following holds: for any $\epsilon$-corruption of the samples, Assumption 3.1 is satisfied. In that event, if $R_0$, $L$, $\lambda$, $\sigma$, and $\epsilon$ are known, then there is a $\mathrm{poly}(n, d, p, \log(1/\delta), \log(R_0/(\sigma\sqrt{\epsilon})))$-time algorithm which produces an estimate $\hat{w}$ satisfying $\|\hat{w} - w^*\|_2 \leq O(\sigma(L^{3/2}/\lambda^2)\sqrt{\epsilon})$ with probability at least $1 - \delta$.*

# 7  Experiments

In this section we corroborate our theory by applying our algorithm ITERATED-GMM-SEVER to several datasets for IV linear regression. See Appendix G for omitted figures and experimental details (e.g. hyperparameter choices and descriptions of the baselines). Error bars are at 25th and 75th percentiles across independent trials.

**Varied Instrument Strength.**  We construct a synthetic dataset with endogenous noise and $1\%$ corruptions, and evaluate our estimator as the instrument strength is varied. Concretely, for dimension

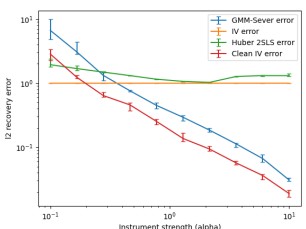
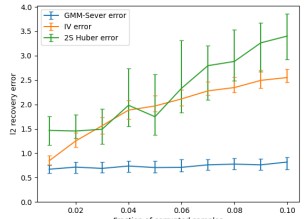
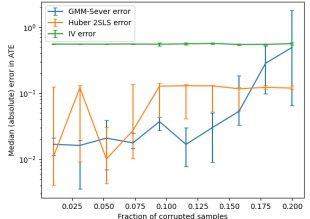

(a) Varied Instrument Strength

(b) Synthetic HE dataset with added corruptions

(c) NLSYM dataset with added corruptions

Figure 1

$d$ and strength $\alpha$, we draw independent samples $(X_i, Y_i, Z_i)_{i=1}^n$ where for unobserved noise $\eta_i \sim N(0, I_d)$, we define instruments $Z_i \sim N(0, I_d)$ and covariates $X_i = \alpha Z_i + \eta_i$, and response $y_i = \langle X_i, \theta^* \rangle + \langle \eta_i, \mathbb{1} \rangle$. For $k = 0.01n$ of the samples, we introduce corruption by setting $Z_i = -A/(k\sqrt{d})$ and $y_i = \sqrt{d}$ where $A = \sum Z_j y_j$, which zeroes out the IV estimate. We take $n = 10^4$, $d = 20$ and $\theta^* = (1, 0, \ldots, 0)$, and vary $\alpha$ from 0.1 to 10. For each $\alpha$, we do 10 independent trials, comparing median $\ell_2$ error of ITERATED-GMM-SEVER with classical IV and two-stage Huber regression. We also compare to the "clean IV" error, i.e. the error of IV on the uncorrupted samples. When $\alpha$ is small, essentially no inference is possible (the clean error is large), but as $\alpha$ increases, our estimator starts to outperform the baselines, and roughly tracks the clean error (Figure 1a). Similar results can be seen for $d = 100$ (Figure 2 in Appendix G.5).

Our next two examples consider *IV linear regression with heterogeneous treatment effects*, a natural setting in which the instruments and covariates are high-dimensional, necessitating dimension-independent robust estimators. Consider a study in which each sample has a vector $X$ of characteristics, a scalar instrument $Z$, a scalar treatment $T$, and a response $Y$. Assuming that the control response and treatment effect are linear in the characteristics, with unknown coefficients $\beta^*$ and $\theta^*$ respectively, and that the response noise is mean-zero conditioned on $Z$ and $X$ (but may correlate with the treatment), we can write the moment conditions

$$\mathbb{E}[XZ(Y - T\langle X, \theta^* \rangle - \langle X, \beta^* \rangle)] = \mathbb{E}[X(Y - T\langle X, \theta^* \rangle - \langle X, \beta^* \rangle)] = 0.$$

This can be interpreted as an IV linear regression with covariates $(TX, X)$ and instruments $(ZX, X)$.

**Synthetic HE dataset.** For parameters $n, d$, we generate a unknown $d$-dimensional parameter vector $\theta^* \sim N(0, I_d)$. We then generate independent samples $(X_i, Y_i, Z_i)_{i=1}^n$ as follows. Draw $X_i \sim N(0, I_d)$ and $Z_i \sim \text{Ber}(1/2)$. The binary treatment is drawn $T_i \sim \text{Ber}(p_i)$ with

$$p_i = \frac{1}{1 + \exp(-Z_i - U_i \bar{X}_i)},$$

where $U_i \sim N(0, 1)$ and $\bar{X}_i = d^{-1/2} \langle X_i, \mathbb{1} \rangle$. Finally, the response is $Y_i = \langle X_i, \theta^* \rangle T_i + \langle X_i, \beta^* \rangle + U_i$ with $\beta^* := 0$.

Ordinary least squares would produce a biased estimate of $(\theta^*, \beta^*)$, since $T\bar{X}$ is correlated with the response noise $U$. However, $U$ is by construction independent of $X$ and $Z$. Thus, in the absence of corruption, IV linear regression with covariates $(TX, X)$, response $Y$, and instrument $(ZX, X)$ should approximately recover the true parameters $(\theta, \beta)$.

For $n = 10^3$ and $d = 20$, the IV estimate still has significant variance, and in this regime, even with no added corruptions, we find that ITERATED-GMM-SEVER has lower recovery error than baselines (Table 1 in Appendix G.5). For $n = 10^4$ and $d = 20$, the IV estimate is more accurate. Hence, we corrupt the first $\epsilon n$ samples, by setting $X_i := \mathbb{1}$ and $Y_i := 3\sqrt{d}$. Varying $\epsilon$ from 0.01 to 0.1, we compute the median $\ell_2$ recovery error of ITERATED-GMM-SEVER, classical IV, and two-stage Huber regression, across 50 independent trials (for each $\epsilon$). The results (Figure 1b) demonstrate that our algorithm is resilient to up to 10% corruptions, whereas both baselines rapidly degrade as $\epsilon$ increases.

**NLSYM dataset.** In this experiment, we use the data of [6] from the National Longitudinal Survey of Young Men for estimating the average treatment effect (ATE) of education on wages. The data

consists of 3010 samples with years of education as the treatment, log wages as the response, and proximity to a 4-year college as the instrument, along with 22 covariates (e.g. geographic indicator variables). For simplicity, we restrict the model to only two covariates (years and squared years of labor force experience) and bias term. We find that the ATE estimated by ITERATED-GMM-SEVER is close to the positive ATE ($\approx 0.277$) estimated by classical IV, suggesting that Card's inference may be robust (Figure 3 in Appendix G.5). Next, we corrupt a random $\epsilon$-fraction of the responses, in a way that negates the ATE inferred by classical IV regression (see Appendix G.2 for method).

Varying $\epsilon$ from 0.01 to 0.2, we perform 10 independent trials (i.e. resampling the subset of corrupted samples each time). For each trial, we compute the ATE estimate of IV regression, the ATE estimate of two-stage Huber regression, and the median ATE estimate of 50 runs of ITERATED-GMM-SEVER. For each $\epsilon$, we then plot the median absolute error of each algorithm across the 10 trials. We find that our algorithm outperforms both baselines, and has lower variance than two-stage Huber regression, up to $\epsilon \approx 0.15$ (Figure 1c; note that error is on log-scale, so the Huber regression is extremely noisy).

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
