# OpenReview forum: "Robust Generalized Method of Moments: A Finite Sample Viewpoint"
_NeurIPS.cc/2022/Conference — NeurIPS 2022 Accept_

### Official Review · Reviewer_g7Q3 · 2022-07-10

**Rating:** 7
**Confidence:** 2
**Soundness:** 3 good
**Presentation:** 3 good
**Contribution:** 3 good

**Summary:**

The SEVER algorithm was proposed in 2019 as a robust algorithm that aims to find an approximate critical point of a collection $f_1,\dots,f_n$ of real-valued functions, with the caveat that a fraction $\epsilon n$ of the sample is corrupted. Here, the authors propose a modification of this algorithm to compute generalized methods of moments estimators: these are estimators of the form $\hat \theta = argmin_{\theta\in \Theta} \|n^{-1}\sum_{i=1}^n g(X_i,\theta)\|^2$. Te output of their algorithm is computationally tractable even in high dimension and can be shown to be robust against a constant fraction of outliers.

The performance of this estimator is tested on both a synthetic on non-synthetic dataset.

**Questions:**

- It is stated l.177 that the filtering algorithm is standard. Then, it is surprising that Lemmas 4.1 and 4.2 are not already found in the literature.
- Lemma 5.1: I think we also need an assumption of the type $\|w_0-w^*\|\leq R_0$ an $R\leq R_0$. (such assumptions are implicitly used in the proof)
- Theorem 5.4: what is $p$?
- In the proof of Lemma A.1: why do we have $\lambda\leq L$? The constants in the three last lines of l.461 seem off: we have $\|w-w^*\|\leq 3R_0$ and not $2R_0$.
- Proof of Lemma 4.1: once again the constants seem off.
- Proof of Lemma 5.2: once again, it looks like we are using that $\|w_0-w^*\|\leq R_0$ an $R\leq R_0$ in the proof of Lemma 5.2 (paragraph l.532)
- **!!!!!!** I do not understand the proof of Theorem 5.4. More specifically, it is stated that "In this event, $|S_t|\geq 2n/3$ for all $t$", and I do not know why this should hold. This is arguably the main theorem in the paper so more details on this proof are needed. (also, how is $X_1$ defined? an ''initialization'' is missing).


Typos:
- l.182: had
- l.478 $\sqrt{L\epsilon}$
- l.511: Step 3 is not defined
- l.518: define $u$
- l.528 third line: $\mathbb{E}_{I_{good}}$

**Limitations:**

Yes.

**Strengths And Weaknesses:**

The paper is well-written and provides a compelling motivation for considering the modified SEVER algorithm: GMM methods can be applied in a wide number of different contexts, so that designing a sound (and computationally tractable) robust method to compute them is a welcome contribution.

From a technical point of view, most results and proofs are modifications of corresponding results in the original SEVER paper. In such, there are no new techniques that are provided.

Although mostly clear, there are several parts throughout sections 4 and 5 where I had trouble following the proofs: there are either small typos with weird numerical constants appearing, or some parts I do not understand (see Questions below). The proofs in Section 6 were not carefully checked.

---

> ### Author Response · Authors · 2022-08-01
> **Author Response**
>
> We thank the reviewer for their time. To address the clarification questions:
>
> 1. [Re. section 4] Yes, we intended that our references on line 178 for the Filter algorithm also applied to Lemmas 4.1 and 4.2. We'll be more explicit about that.
>
> 2. [Re. Lemmas 5.1 and 5.2] You're right, the assumption in Theorem 5.4 that $B_R(w_0)$ is contained in $B_{2R_0}(0)$ should also be stated as a condition in Lemmas 5.1, 5.2, and 5.3. Thanks for pointing this out.
>
> 3. [Re. $p$] $p$ is the number of moment constraints; see Line 155.
>
> 4. [Re. the relation between $\lambda$ and $L$] This is because $\lambda$ lower bounds the smallest singular value of the expected gradient matrix, and $L$ upper bounds the largest singular value of the same matrix (see lines 159-160 in Assumption 3.1; the upper bound can be deduced by an application of Jensen's inequality)
>
> 6. [Re. Proof of Theorem 5.4] This is by an induction on $t$. We know that $10\epsilon n < n/3$. Suppose there is some $t$ such that $|S_t| < 2n/3$, and pick the minimal such $t$. Then $|S_{t-1}| \geq 2n/3$. So by definition, $X_t$ is the size of the symmetric difference between $S_t$ and $I_\text{good}$. So $|S_t| \geq |I_\text{good}| - X_t \geq (1-\epsilon)n - 9\epsilon n \geq 2n/3$. Contradiction, so $|S_t| \geq 2n/3$ for all t.
>
>     Thanks for bringing to our attention that this argument is non-obvious; we'll write it out explicitly in the paper. $X_1$ is defined as $|S_1 \triangle I_{good}|.$
>
> 7. [Re. constants and other typos] Thanks for catching these. You're right about the constants in line 461 and proof of Lemma 4.1. In Lemma 5.1, "Step 3" should be "Step 6", and $u$ is defined on line 5 of Algorithm 2.

---

> > ### Comment · Reviewer_g7Q3 · 2022-08-09
> > **Response**
> >
> > Thank you for the clarification.

---

### Official Review · Reviewer_xqsD · 2022-07-11

**Rating:** 7
**Confidence:** 3
**Soundness:** 4 excellent
**Presentation:** 4 excellent
**Contribution:** 3 good

**Summary:**

* In this paper an algorithm and its theoretical guarantees are given for the problem of Generalized Method of Moments, in the setting
where an $\epsilon$ fraction of data samples could be adversarially corrupted.
* The guarantees are given under deterministic assumptions on the uncorrupted part of the data and on the moment function g.
* It is then proved that samples from the instrumental variables linear and logistic regression models satisfy the above assumptions with high probability.

* The arguments build upon series of recent work on robust estimation of means and local optima. However, the contributions presented here on top of these ideas are considerable.

* It is stated that the methods presented in this paper are computationally tractable and, moreover, efficient, in contrast to existing work on Robust GMM.

**Questions:**

I will be glad to see the author's comments on points 1 and 2 above.

**Limitations:**

The assumptions made in the paper were appropriately discussed.

**Strengths And Weaknesses:**

As discussed above, I believe this is a well written paper with a solid contribution.

The main weakness in my view is that eventual sample complexity $n$ of the instrumental variables algorithms in Section 6 is
$(d+p)^5$, where $d$ is the dimension of the features and $p$ the dimension of the instrument variable. These clearly are infeasible for all
but very small d and p.

The synthetic data experiments are performed with $n$ $d$ and $p$ that do not satisfy these bounds, and results indicate that perhaps the bound may be strengthened. The experiment with NLSYM data is performed with d=2, instead of d=22 in the original data.


1) Since the bounds in the paper are not computationally feasible, a deeper comparison to existing methods should be performed.
   Why the present work is better than sources [20],[11], which are referred to as computationally intractable in the paper?
2) What would be the results of the NLSYM experiment if it was performed with full d=22 data?

---

> ### Author Response · Authors · 2022-08-01
> **Author Response**
>
> We thank the reviewer for their time. To address their questions:
>
> 1. While our provable sample complexity bounds (as is typical of worst-case theoretical guarantees) are loose compared to what is needed in practice, the critical feature of our results is that the sample complexity and runtime are *polynomial*, in contrast with prior algorithms which required exponential time, and could not even be implemented without resorting to heuristics. We also observe that our algorithm has provable guarantees whenever the given dataset satisfies the (deterministic) Assumption 3.1, which may hold even if the number of samples is far less than what 6.2 and 6.3 suggest.
>
>     We also expect that parts of our bounds (in particular, the dependence on $d$ and $p$) could potentially be improved. Such an improvement was obtained e.g. for robust linear regression, where the Sever paper (which requires $d^5$ samples) was followed by [CATJFB '20] ''Optimal Robust Linear Regression in Nearly Linear Time", which uses a similar framework (with additional algorithmic insights) to achieve nearly linear time and sample complexity.
>
>     Our paper can be seen as taking the crucial first step of breaking the exponential barrier for the more general robust GMM setting.
>
> 2. Good question; unfortunately we found that (even without added corruption), IV with heterogeneous treatment effects is wildly unstable to adding/subtracting a variable on the whole d=22 dataset, probably due to weak instruments; hence, we used a simplified model with only 2 covariates to be sure that when adding corruption, we were starting from a stable baseline. In hindsight, we believe that Card's original analysis may have used IV with homogeneous treatment effects, a simpler model which is likely more stable but makes stronger statistical assumptions. Of course, our algorithm applies to that model as well, but we did not conduct that particular experiment.

---

### Official Review · Reviewer_JzqV · 2022-07-13

**Rating:** 6
**Confidence:** 4
**Soundness:** 3 good
**Presentation:** 3 good
**Contribution:** 3 good

**Summary:**

The paper studies the problem of robust inference when the loss is taken as the norm of generalized moments. The authors provide robust GMM algorithm based on the SEVER algorithm and the iterative filtering framework. The robust GMM algorithm filters both the moments as well as the directional derivatives of the moments.

**Questions:**

1. I'd like the authors to provide more comment on the optimality of the results, in particular for Theorem 5.5, 6.2 and 6.3. It seems to me that in the special case of robust mean estimation with
$g_i(w) = X_i - w$, Assumption 3.1 holds when X_i has bounded covariance and Theorem 5.5 gives a tight bound except for a worse breakdown point compared to [1] (and the algorithm basically coincides with the iterative filtering for mean estimation). If it is correct, perhaps it's worth mentioning after Theorem 5.5.

2. However, I'm unsure about the optimality of Theorem 6.2 and 6.3, in particular, for a strongly convex function with bounded variance, the parameter estimation error is usually $O(\epsilon)$, see e.g. [2]. And the rate becomes better with higher moment assumption. However, the rate under bounded 8th moment is still $O(\sqrt{\epsilon})$. Can the authors comment on whether this rate is tight or not? Also, would strongly convexity or any other property helps improve the rate in Theorem 5.5?

[1] Zhu, Banghua, Jiantao Jiao, and Jacob Steinhardt. "Robust estimation via generalized quasi-gradients." Information and Inference: A Journal of the IMA 11.2 (2022): 581-636.

[2] Yin, Dong, et al. "Byzantine-robust distributed learning: Towards optimal statistical rates." International Conference on Machine Learning. PMLR, 2018.


**Strengths And Weaknesses:**

The authors provide a new algorithm, robust GMM, based on SEVER and iterative filtering. Theoretical guarantee is provided for the proposed algorithm. The authors also provide concrete applications for robust IV linear regression and logistic regression. I enjoy reading the paper and understanding the techniques. My main concern is about the optimality of the rates, as detailed in the question section.

---

> ### Author Response · Authors · 2022-08-01
> **Author Response**
>
> We thank the reviewer for their time.
>
> 1. That is correct, the $O(\sqrt{\epsilon})$ rate matches that of robust mean estimation under bounded covariance, where $O(\sqrt{\epsilon})$ is optimal (see e.g. [DKKLMS 17] "Being Robust (in High Dimensions) Can Be Practical").
>
> 2. Note that IV regression generalizes ordinary linear regression. Under standard robust regression assumptions, $O(\sqrt{\epsilon})$ is optimal (see Theorem D.2 in [CATJFB '20] "Optimal Robust Linear Regression in Nearly Linear Time"). Our assumption for robust IV regression (Assumption 6.1) is the same as for ordinary regression, except that we also require bounded 8th moments. Given that, we would expect that a slight improvement on $O(\sqrt{\epsilon})$ is possible.
>
>     The 8th moment bound is a technical requirement that we only need in one place: Lemma F.6, to prove concentration of 4th moments via a Bernstein bound. It would be interesting if this could be eliminated.
>
> 3. [*Re. strong convexity*] For the simpler setting of robust function minimization, assuming the (average uncorrupted) function is strongly convex helps with prediction error but does not seem to help with parameter recovery error (see Corollary B.4 in [DKKLMS '19] "Sever: A Robust Meta-Algorithm for Stochastic Optimization"). We might expect a similar phenomenon here, although it's not as clear what we would assume strong convexity of.

---

### Official Review · Reviewer_rQ7k · 2022-07-15

**Rating:** 5
**Confidence:** 3
**Soundness:** 3 good
**Presentation:** 3 good
**Contribution:** 2 fair

**Summary:**

In this paper, the authors claim to develop the first computationally efficient Generalized Method of Moments (GMM) estimator that is robust to a constant fraction of arbitrary outliers. The authors instantiate this estimator for two important special cases of GMM, namely instrumental variable (IV) linear regression and IV logistic regression, under distributional assumptions about the covariates, instruments, and responses. Some numerical results are provided to support the theory.

**Questions:**

Some minor suggestions are listed as follows:
1. The authors should provide the full name of the SEVER algorithm (at least when it first appears on Page 2).
2. The figures in Figure 1 look quite vague. Perhaps the authors can present them as separate figures (i.e., Figures 1, 2, 3 instead of Figures 1a, 1b, 1c).

**Limitations:**

Not applicable.

**Strengths And Weaknesses:**

Strengths:

1. A computationally efficient and robust GMM estimator should be of interest to the community.
2. This paper is generally well-written and the theoretical results seem to be reasonable and reliable.

Weaknesses:

1. The algorithmic and theoretical contributions in this work seem to be incremental to me. In particular, the authors mention that "Estimators with the above properties have been developed for various fundamental high-dimensional problems, including mean and covariance estimation [6, 8], linear regression [9, 2], and stochastic optimization [25, 7]"  and that the algorithm proposed in this work is "a simple modification to the SEVER algorithm for robust stochastic optimization [7]". I appreciate the authors' effort to provide the $\textbf{Technical Overview}$. But I hope that the authors can demonstrate more clearly what are the major technical novelty in this submission (especially compared to [7]).

2. Personally, I do not like Assumption 3.1. It involves too many conditions. Even if the authors show in Theorems 6.2 and 6.3 that this assumption is satisfied by robust IV Linear/Logistic Regression, there are still too many involved conditions/parameters that make this assumption not easy to parse. I guess for Algorithms 2, 3, and 4, the situation is even worse. That is, there are too many input parameters for these algorithms, and for some of the parameters (e.g., the noise level $\sigma^2$ and well-conditionedness parameters $\lambda$ and $L$), I am not sure whether it is reasonable to use them as the inputs of the algorithms. In my opinion, practitioners typically prefer simple algorithms. But the algorithms provided in this submission seem to be too complicated (so many parameters!) for practitioners.

3. The experimental results are not very convincing. In particular, Figures 1a and 1b are for synthetic data. Figure 1c is for real data, which is more desired, but it looks quite messy. For some cases, Huber regression outperforms the method provided in this work. In addition, the authors mention that "note that error is on log-scale, so the Huber regression is extremely noisy". It seems that the method proposed in this work is also quite noisy.

4. Some references are missing. For example, the authors should add references for the sentence "Unfortunately, like most other classical estimators in statistics, the GMM estimator suffers from a lack of robustness: a single outlier in the observations can arbitrarily corrupt the estimate." and the sentence "However, practitioners in econometrics and applied statistics often employ more sophisticated inference methods such as GMM and IV regression." (Since compared to prior works, IV regression is emphasized in this submission and I am not familiar with IV regression. I am curious about how important IV regression is in practice). In the Experiments section, the authors should provide references to the baseline methods.

---

> ### Author Response · Authors · 2022-08-01
> **Author Response**
>
> We thank the reviewer for their time.
>
> 1. [Re. ''technical novelty"] As the reviewer's quotes highlight, we openly acknowledge that our algorithm is (in hindsight) a technically simple modification of Sever. Moreover, we do not see why this is a weakness. Indeed, the reviewer observes that ''practitioners typically prefer simple algorithms".
>
> 2. [Re. the number of hyperparameters] See Appendix G.1 of the supplementary material. For our actual experiments we find that parametrizing the algorithm with only two hyperparameters is sufficient. Moreover, we did not need to make serious attempts at tuning the hyperparameters; indeed, we conducted additional experiments verifying that the algorithm performance is stable to the choice of hyperparameters (Appendix G.3). Thus, while the theoretical analysis involves many parameters, the algorithm as-implemented is actually easy to use.
>
> 3. [Re. Fig 1c] Because the error is on a log-scale, the apparent length of the error bar is not indicative of the variance of the estimator error. We say that the Huber regression is noisy because the top error bar (i.e. 75th percentile) exceeds $\approx 0.14$ for all but one setting of epsilon. In contrast, up to 15\% corruptions, our algorithm's 75th percentile of error is never more than $0.04$, and usually around $0.02$ to $0.03$. The apparent "noise" in our algorithm's error bars is mostly indicative of how on many trials, our algorithm achieves very *low* error ($\leq 0.01$), an order of magnitude better than the Huber regression.
>
> 4. The fragility of the GMM estimator to even a single gross outlier is simply because it generalizes e.g. ordinary least squares, where this behavior is well-known (see e.g. [Huber 1973] ''Robust regression: asymptotics, conjectures and Monte Carlo''). For a reference on the importance of IV regression in practice, see e.g. ''Instrumental variables and the search for identification: From supply and demand to natural experiments" (Angrist and Krueger, 2001).
>
> References for the baseline algorithms are provided in the supplementary experimental details (Appendix G.1).

---

### Meta-Review · Area_Chair_2Lwa · 2022-08-26

**Recommendation:** Accept
**Confidence:** Less certain

**Metareview:**

The authors proposed new computationally efficient Generalized Method of Moments (GMM) estimators that are robust to a constant fraction of arbitrary outliers. It is based on modifications of existing algorithms such as SEVER and filtering. Although the analysis is not tight in all the cases, this paper presents an interesting first step towards solving this general family of problems.


**Award:**

No

---

### Decision · Program_Chairs · 2022-09-14

Accept